# Characterization of Sol-Gel Derived Calcium Hydroxyapatite Coatings Fabricated on Patterned Rough Stainless Steel Surface

**Vilma Jonauske [1],\*, Sandra Stanionyte [2], Shih-Wen Chen [3], Aleksej Zarkov [1], Remigijus Juskenas [2], Algirdas Selskis [2], Tadas Matijosius [2], Thomas C. K. Yang [3], Kunio Ishikawa [4], Rimantas Ramanauskas [2] and Aivaras Kareiva [1]**

[1] Department of Inorganic Chemistry, Faculty of Chemistry and Geosciences, Vilnius University, Naugarduko str. 24, LT-03225 Vilnius, Lithuania; aleksej.zarkov@chf.vu.lt (A.Z.); aivaras.kareiva@chgf.vu.lt (A.K.)

[2] Center for Physical Sciences and Technology, Sauletekio av. 3, LT-10257 Vilnius, Lithuania; sandra.stanionyte@ftmc.lt (S.S.); remigijus.juskenas@ftmc.lt (R.J.); algirdas.selskis@ftmc.lt (A.S.); tadas.matijosius@ftmc.lt (T.M.); rimantas.ramanauskas@ftmc.lt (R.R.)

[3] Department of Chemical Engineering and Biotechnology, National Taipei University of Technology, 1, Sec. 3, Chung-Hsiao E. Road, Taipei 106, Taiwan; shihwenc@gmail.com (S.-W.C.); ckyang@mail.ntut.edu.tw (T.C.K.Y.)

[4] Department of Biomaterials, Faculty of Dental Science, Kyushu University, Maidashi, Higashi-Ku, Fukuoka 8120053, Japan; ishikawa@dent.kyushu-u.ac.jp

\* Correspondence: vilma.ciuvasovaite@gmail.com; Tel.: +370-6-106-4808

**Abstract:** Sol-gel derived calcium hydroxyapatite ($Ca_{10}(PO_4)_6(OH)_2$; CHA) thin films were deposited on stainless steel substrates with transverse and longitudinal patterned roughness employing a spin-coating technique. Each layer in the preparation of CHA multilayers was separately annealed at 850 °C in air. Fabricated CHA coatings were placed in simulated body fluid (SBF) for 2, 3, and 4 weeks and investigated after withdrawal. For the evaluation of obtained and treated with SBF coatings, diffuse reflectance infrared Fourier transform spectroscopy (DRIFTS), X-ray diffraction (XRD) analysis, Raman spectroscopy, XPS spectroscopy, scanning electron microscopy (SEM) analysis, and contact angle measurements were used. The tribological properties of the CHA coatings were also investigated in this study.

**Keywords:** calcium hydroxyapatite; sol-gel synthesis; thin films; spin coating; surface roughness; simulated body fluid

## 1. Introduction

For many years, 316L stainless steel has been widely used for orthopedic and orthodontic applications due to its excellent mechanical properties, good biocompatibility, and high corrosion resistance [1,2]. Metallic implants themselves are not bioactive; therefore, medical doctors and scientists are always concerned with achieving acceptable integration of implants into living bone. The main problem with metallic implants is the possible formation of fibrous tissue around the guest body implanted into bone [3]. Fibrous tissue formation is caused by the absorption of protein and other organic molecules on the hydrophobic surface of metals. Hydrophobic surfaces attract these organic molecules with enhanced formation of the above-mentioned biofilms [4,5]. It was suggested that formation of protein capsules may cause inflammation at the implant-bone interface, followed by

the rejection of an implant [6]. Therefore, modified surfaces with antibacterial properties have been developed [7–9].

Implants are foreign bodies, and therefore, surface morphology and other properties have an impact on the behavior of bone cells that come into contact with implants [10,11]. Of course, the immune system of an individual patient is very important for the response to implanting biomaterials [12]. Bioglasses and other bioceramics such as calcium phosphates bond to natural bone and support the proliferation of new cells [2,13–15]. Surface wettability enhances osteoblast adhesion of human cells and also cell proliferation [16–18]. The problems associated with the application of metallic implants might be solved by the formation of calcium hydroxyapatite ($Ca_{10}(PO_4)_6(OH)_2$; CHA) on the surface [19,20]. CHA is a well-known biomaterial for its bioactivity, biocompatibility, and ability to induce growth of a new bone tissue. After implantation of a ceramic-coated, metal implant, the CHA forms strong chemical bonds with natural bone tissue due to its structural and chemical similarity to the bone [21,22].

Application of biphasic calcium phosphate in biocomposites or hydroxyapatite showed no fibrous encapsulation of the particles and composites after implantation allowing further infiltration of cells within the sample implants [23–25].

Previously we described sol-gel chemistry processing for the preparation of calcium hydroxyapatite films [26–28]. The CHA thin films were fabricated on substrates including silicon, quartz, and modified titanium. Both dip-coating and spin-coating techniques were compared. The same sol-gel method was used for the synthesis of CHA coatings on stainless steel substrates [29]. The quality of thin films obtained using the spin-coating technique was slightly better than when using dip-coating. However, the adhesion of CHA coatings obtained by both processes were less than desired. Recently, the effects of surface roughness on the adhesive properties of thin films were discussed and investigated [30]. In our study, the CHA thin films were applied to roughened stainless steel substrates using the spin-coating technique. Transverse and longitudinal roughness was patterned on the metal. The bioactivity of sol-gel derived calcium hydroxyapatite coatings in simulated body fluid are presented.

## 2. Experimental

For the sol-gel processing, calcium acetate monohydrate ($Ca(CH_3COO)_2 \cdot H_2O$; 99.9%; Fluka, Saint Louis, MO, USA, 5.2854 g (0.03 mol)) was dissolved in 40 mL of distilled water and mixed with 4 mL of 1,2-ethanediol (99.0%; Alfa Aesar, Haverhill, MA, USA). Then, a mixture of ethylenediaminetetraacetic acid (EDTA; 99.0%; Alfa Aesar, Haverhill, MA, USA, 9.6439 g (0.033 mol)) and triethanolamine (99.0%; Merck, Darmstadt, Germany, 24 mL) was added to the solution, which was continuously stirred for 10 h. Next, phosphoric acid ($H_3PO_4$; 85.0%; Reachem, Manhattan, NY, USA, 1.23 mL) was added with stirring for about 24 h. Finally, the obtained sol-gel solution was mixed with 3% solution of polyvinyl alcohol (PVA7200, 99.5%; Aldrich, Saint Louis, MA, USA) in the proportion 25 to 15 mL. This solution was used as sol-gel precursor solution for the synthesis of CHA coatings on prepared 316L stainless steel substrate. The surface of square plates of 10 mm × 10 mm × 0.5 mm was modified by formation of transverse and longitudinal patterned roughness (less than 0.2 µm) prior to the spin-coating of the Ca–P–O sol-gel solution (0.5 mL). A sandpaper (2500 grit) was used to obtain the pattern. Two pieces of sandpaper were lightly rubbed against each other to remove any possible large particles to protect the surface from gouging. Then the metal was roughened with the sandpaper applying medium pressure from an index finger.in two directions 90° apart. A schematic diagram of the preparation of the CHA coatings is presented in Figure 1.

The spinning procedure was performed for 1 min in air with a spinning rate of 2000 revolutions per minute. Each layer in the preparation of CHA multilayers was separately annealed at 850 °C for 5 h in air. CHA thin films with 15, 20, 25, and 30 layers were fabricated. To evaluate in vivo bioactivity of the sol-gel derived CHA coatings, the samples were immersed into simulated body fluid (SBF) for 2, 3, and 4 weeks. The SBF was prepared using the method proposed by Kokuba and Takadama [3]. The concentrations of ions in the prepared SBF solution are presented in Table 1.

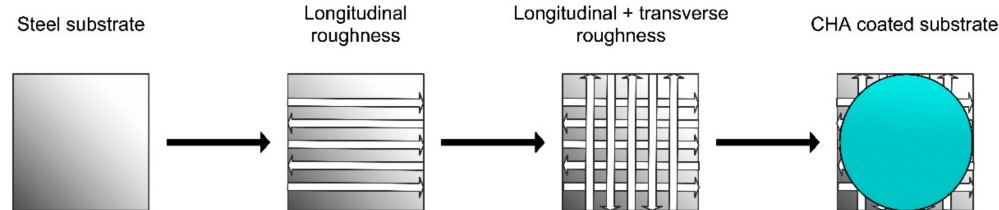

**Figure 1.** A schematic diagram of the preparation of CHA coatings.

**Table 1.** Nominal ion concentrations of simulated body fluid (SBF) in comparison with those in human blood plasma.

| Ion | Ion Concentration (mM) | |
|---|---|---|
| | **Blood Plasma** | **SBF** |
| $Na^+$ | 142.0 | 142.0 |
| $K^+$ | 5.0 | 5.0 |
| $Mg^{2+}$ | 1.5 | 1.5 |
| $Ca^{2+}$ | 2.5 | 2.5 |
| $Cl^-$ | 103.0 | 147.8 |
| $HCO_3^-$ | 27.0 | 4.2 |
| $HPO_4^{2-}$ | 1.0 | 1.0 |
| $SO_4^{2-}$ | 0.5 | 0.5 |
| pH | 7.2–7.4 | 7.40 |

X-ray diffraction (XRD) analysis was performed with a D8 Focus diffractometer (Bruker AXS Inc., Hamburg, Germany) with a LynxEye detector (Bruker, Billerica, MA, USA) using Cu Kα radiation. The changes in the layers were evaluated using diffuse reflectance infrared Fourier transform spectroscopy (DRIFTS) with a FTIR Spectrum BX II spectrometer (Perkin–Elmer, Waltham, MA, USA). The microstructure and morphology of the obtained samples were investigated using a SU-70 scanning electron microscope (SEM) (Hitachi, Tokyo, Japan). For the evaluation of the hydrophobicity of the CHA coatings, the contact angle measurements were performed using contact angle meter (CAM) (CAM 100, KSV, San Francisco, CA, USA). Distilled water was used for CAM measurements. Raman spectra were measured on Ramboss 500i micro spectrometer (Dongwo, Hsinchu, Taiwan). XPS measurements of the synthesized samples were carried out employing the JPS-9030 spectrometer (JEOL, Akishima, Japan). For tribological measurements, a TriTec SA CSM Tribometer (Anton Paar, Buchs, Switzerland) was used with a ball-on-plate linearly reciprocal configuration, as described by Matijosius et al. [31]. A corundum ball (RGP International Srl, Cinisello Balsamo MI, Italy) of 6 mm (outer diameter) was held stationary, and the load was 1 or 5 N. Stainless steel substrates coated with calcium hydroxyapatite were used as the moving part, which was mounted on a pre-installed tribometer module. Linear reciprocal motion within the amplitude of 2 mm was maintained resulting in a 4 mm range and 8 mm of the total distance for one reciprocal friction cycle. At a velocity of 2 cm/s, each friction cycle produced approx. 100 data points of the 'instantaneous' coefficient of friction (COF).

## 3. Results and Discussion

Figure 2 shows XRD patterns of CHA films synthesized using the spin-coating technique on a steel surface with patterned roughness.

The multilayer coatings were obtained by increasing the number of spinning and annealing cycles from 15 to 30. The dependence of CHA formation on the times of spinning and annealing is evident. With 15 spinning and annealing procedures, iron oxides ($Fe_2O_3$ and $Fe_{2.932}O_4$) were the dominant crystalline phases. The iron oxide phase was formed by heating an uncoated but roughened stainless steel substrate at the same temperature (see Figure 3).

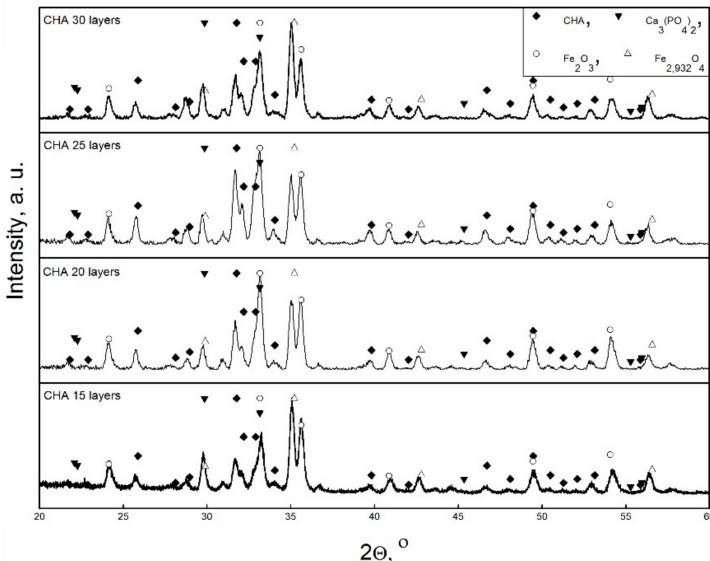

**Figure 2.** XRD patterns of calcium hydroxyapatite (CHA) coatings fabricated using different times of spinning and annealing procedures.

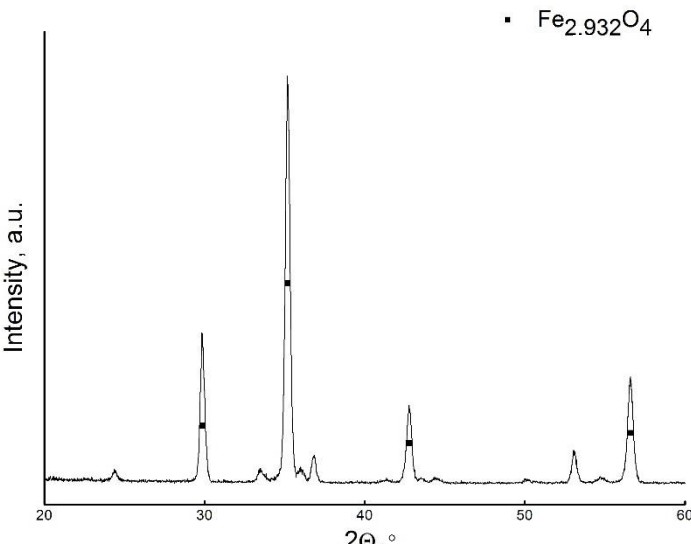

**Figure 3.** XRD pattern of uncoated rough 316 L stainless steel substrate annealed at 850 °C. $Fe_{2.932}O_4$ is the only dominant phase.

Even after 15 coating cycles, the calcium phosphate phases such as CHA and $Ca_3(PO_4)_2$ have also formed. The intensity of diffraction lines attributable to the CHA phase monotonically increased with increasing the number of layers to 20 and 25. Well resolved CHA diffraction peaks were seen and in the XRD pattern of the sample with 30 layers. However, after 30 spin-coating procedures the intensity of peaks attributable to calcium phosphate phases surprisingly decreased. These XRD results allow us to draw the initial conclusion that the most crystalline CHA sample was obtained with 25 CHA layers.

The FTIR spectra with the DRIFTS attachment are presented in Figure 4 for the CHA coatings.

The FTIR spectra were almost identical among all the samples with various number of layers. The characteristic calcium hydroxyapatite absorption bands in the range of 1100–950 $cm^{-1}$ were clearly visible [32]. Strong bands located at 1035 and 1090 $cm^{-1}$ resulted from the $\nu_1$ symmetric P–O stretching vibrations in $PO_4^{3-}$ [32,33]. The broad band visible in the range of 3600–3300 $cm^{-1}$ is associated with O–H stretch vibrations and attributed to adsorbed water [29,32,34]. In all spectra, weak and broad bands in the range of 1550–1370 $cm^{-1}$ were also seen. The origin of these bands is related

to the stretching and bending modes of $CO_3^{2-}$ in CHA (C–O bond), despite the last band being slightly shifted to the region of lower wavenumbers. The existence of low-intensity bands at about 870 cm$^{-1}$ may be ascribed to the $\nu_2$ bending mode of $CO_3^{2-}$ (C–O bond) in CHA and confirmed the presence of carbonated apatite in the samples [32]. No specific bands attributable to oxyhydroxyapatite $Ca_{10}(PO_4)_6(OH)_{2-2x}O_x$ were detected in our spectra; however, such bands were present for the CHA thin films deposited on Si substrate [35].

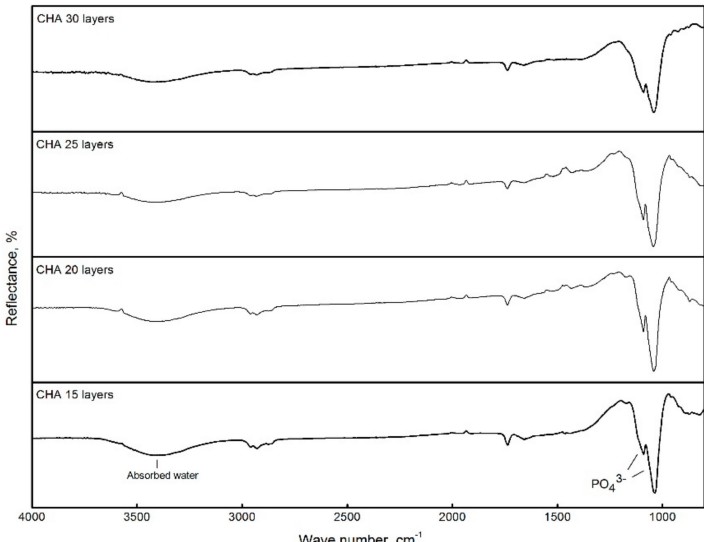

**Figure 4.** FTIR spectra of sol-gel derived films with 15, 20, 25, 30 layers of CHA.

Figure 5 shows Raman spectra in the 200–1400 cm$^{-1}$ spectral region of the CHA samples deposited on rough stainless steel substrate.

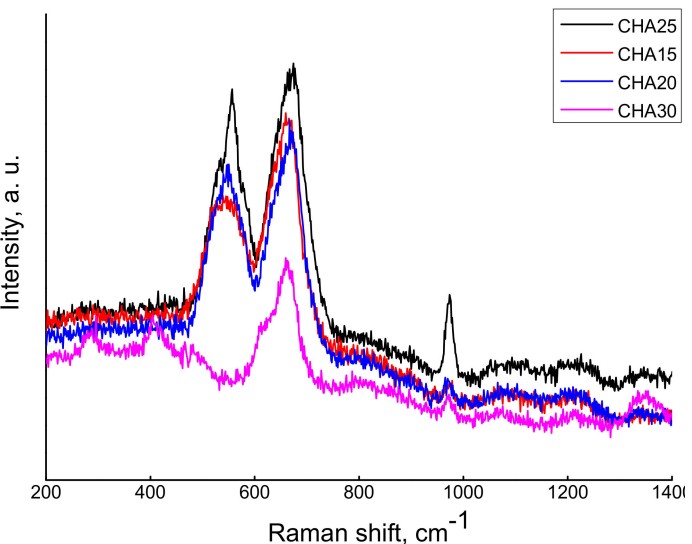

**Figure 5.** Raman spectra of the CHA samples containing 15, 20, 25, and 30 layers of Ca–P–O gel deposited on rough stainless-steel substrate in 200–1400 cm$^{-1}$ spectral region.

The most intensive Raman bands were observed for the CHA sample having 25 layers. The intensity of the bands monotonically increased with the number of layers but decreased for the specimen having 30 layers. The most intense bands were observed at 580 and 640 cm$^{-1}$. These bands could be assigned to the triple degenerate (F$_2$ symmetry) asymmetric bending modes $\nu_4$ of phosphate group in calcium hydroxyapatite [36]. Interestingly, the band at 580 cm$^{-1}$ was not observed in Raman spectrum of the

CHA sample synthesized with 30 layers. The band at about 960 cm$^{-1}$ is due to $\nu_1$ (A$_1$) symmetric stretching vibration of the tetrahedral PO$_4{}^{3-}$ group. This band was the most intense in the Raman spectrum of the CHA sample synthesized on a silicon nitride substrate [35]. For the specimens in this study, the intensity of this band was substantially less. The peak position of this band confirms the exact stoichiometry (molar ratio Ca:P = 1.667) for calcium hydroxyapatite [37,38]. The Raman spectroscopy results are in good agreement with the XRD analysis data, because in both spectra the most intense peaks attributable to CHA were visible for the sample with 25 layers.

The XPS of samples was measured by calibrating the binding energy scale with C 1*s* peak as reference. All XPS survey spectra (Figure 6a) exhibited signals characteristic of calcium, oxygen, phosphorous, and carbon. The high-resolution O 1*s* XPS spectra (Figure 6b) showed a signal centered at 535.1 eV corresponding to O–P–O and OH bonding in hydroxyapatite [39]. Figure 6c shows P 2*p* XPS spectra, which consist of 2*p*$^{3/2}$ and 2*p*$^{1/2}$ components, corresponding to O–P–O bonding in the (PO$_4$) network of calcium hydroxyapatite and tricalcium phosphate (TCP) [40]. The high resolution Ca 2*p* spectra are shown in Figure 6d. That spectra consists of two peaks at 351.1 and 354.7 eV. These peaks are attributed to the spin-orbit splitting components of 2*p*$^{3/2}$ and 2*p*$^{1/2}$ with energy difference of 3.4 eV [41]. The elemental analysis, represented as Ca to P ratio, showed that the Ca to P ratio was lower than in stoichiometric calcium hydroxyapatite (1.67). This is not surprising, since the samples also contain TCP. The Ca:P varied between 1.40 and 1.51 depending on the number of sol-gel layers.

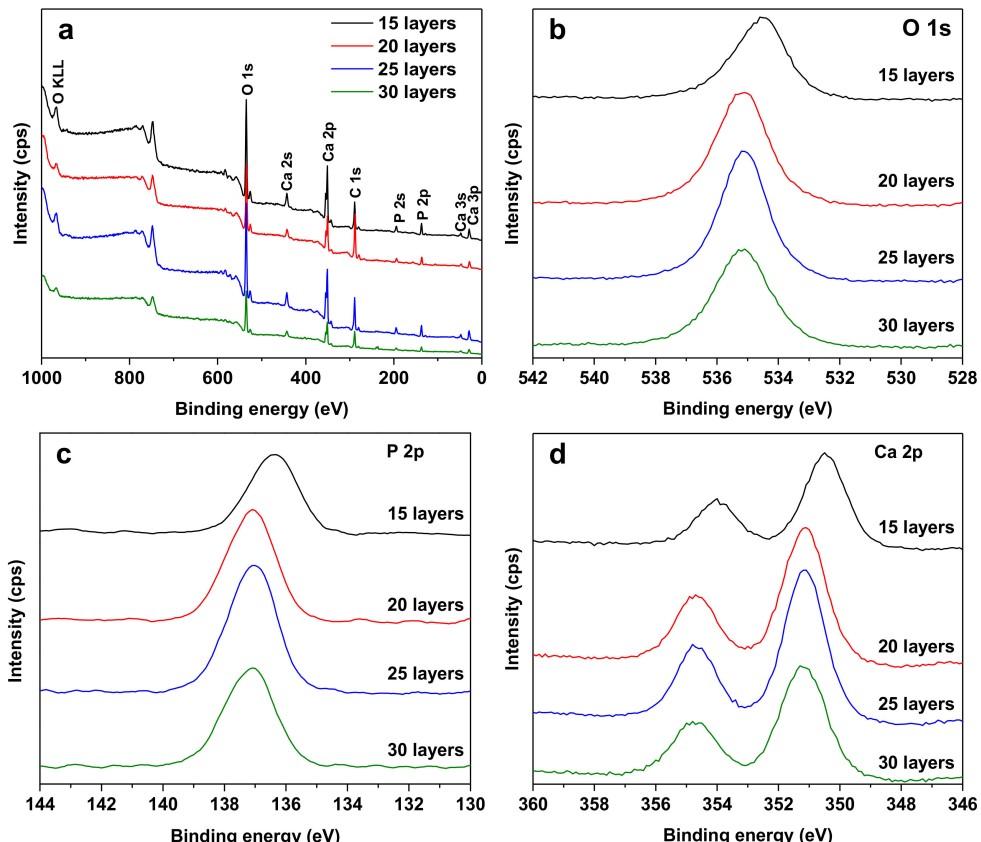

**Figure 6.** XPS survey spectra (**a**) and high-resolution O 1*s* (**b**), P 2*p* (**c**) and Ca 2*p* (**d**) XPS spectra taken from the surface of deposited and annealed CHA films with different numbers of layers.

The SEM micrographs of the surfaces of obtained CHA samples are shown in Figure 7.

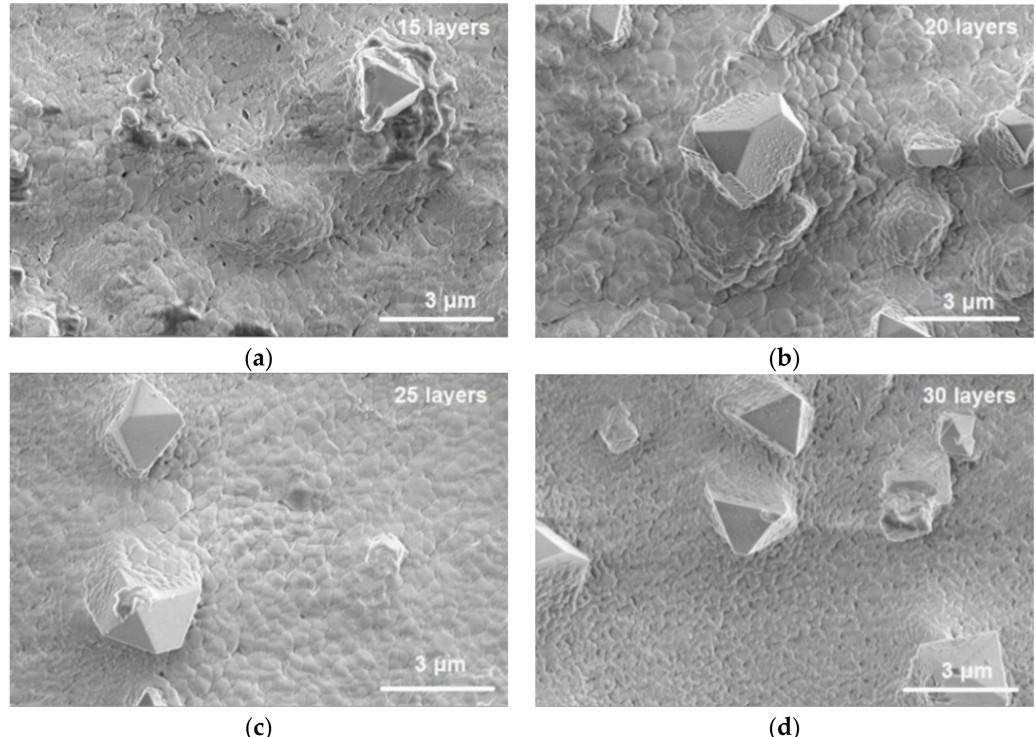

**Figure 7.** SEM micrographs of the CHA samples containing (**a**) 15, (**b**) 20, (**c**) 25, and (**d**) 30 layers. Magnification 15,000.

The surface of the specimen obtained after 15 dipping times is uneven with clearly pronounced asperities and pores. The surface smoothness was better when the number of layers was 20. The steel substrate after 25 spin-coating procedures was even more uniform. The surface with 25 layers of CHA was composed of homogeneously distributed well interconnected spherical grains about 250 nm in size. The layer was continuous and pore-free. Surprisingly, the morphological features of the sample with 30 layers changed, and the formation of nanospheres on the surface cannot be seen. Nano-sized pores were visible in the sample containing 30 layers of CHA. The irregularly shaped crystals are probably iron oxides. The results of the tribological measurements are presented in Figure 8. COF changes were observed after more loading cycles were performed for either a 1 or 5 N load.

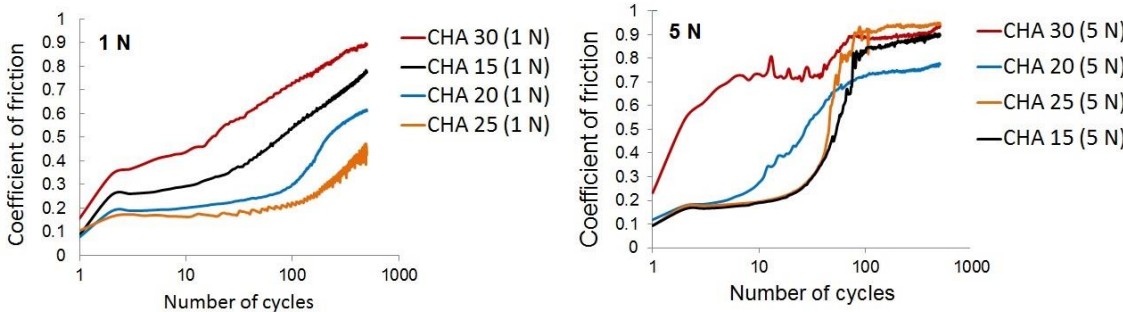

**Figure 8.** Friction dependence on the coating cycles of CHA layers deposited on the patterned rough surface of stainless steel substrate under (**a**) 1 and (**b**) 5 N loads. Number of layers: 15 layers—CHA 15, 20 layers—CHA 20, 25 layers—CHA 25, and 30 layers—CHA 30.

In most samples, the instantaneous COF values remained similar throughout the selected range of the reciprocal motion, i.e., the middle 80% of a segment. Tribological effectiveness (lowest COF) of sol-gel derived CHA layers coated on the rough stainless steel substrates was evaluated under 1 and 5 N loads. The layer thickness of CHA influenced friction from 15 to 30. Layers of 15 and 30

had the lowest friction by showing a gradual increase of COF after a few friction cycles under 1 N load. Samples with 30 coating cycles had the highest friction. This effect may be explained by rapid CHA layer deformation and degradation, producing wear debris particles that damaged the substrate. The CHA coating with 30 layers was thick enough to break off or delaminate into debris particles. Another possible mechanism to explain such an increase in friction could be related to changes in the surface morphology of these CHA specimens. The best tribological effectiveness (lowest COF) was CHACOF reduction below 0.2 for 100 friction cycles under 1 N load. Friction for the sample with 30 CHA layers evaluated under 5 N load was even higher; the COF value was 0.6 after just a few friction cycles. The samples with 15 and 25 layers had much lower COF, below 0.2 for 30 friction cycles. Despite lower durability (30 vs. 100 friction cycles) the CHA layers on the patterned surface resist abrasion under considerable loads. Thin layers, produced after 15 coating cycles, were tribologically effective under 5 N load, while under 1 N load this effect was not observed. Overall, CHA layers showed the best protective and anti-frictional properties when produced after 25 coating cycles under 1 and 5 N loads. SEM micrographs show that the steel substrate with 25 layers was evenly coated with homogeneously distributed spherical grains of about 250 nm in size.

The contact angle measured for the uncoated patterned stainless-steel substrate was 81.3°, as shown in Figure 9.

The contact angle values of all CHA coatings fell within a small range (113°–116°). The images of water drops on CHA surfaces coated with 15, 20, 25, or 30 sol-gel layers show the CHA coatings were slightly hydrophobic.

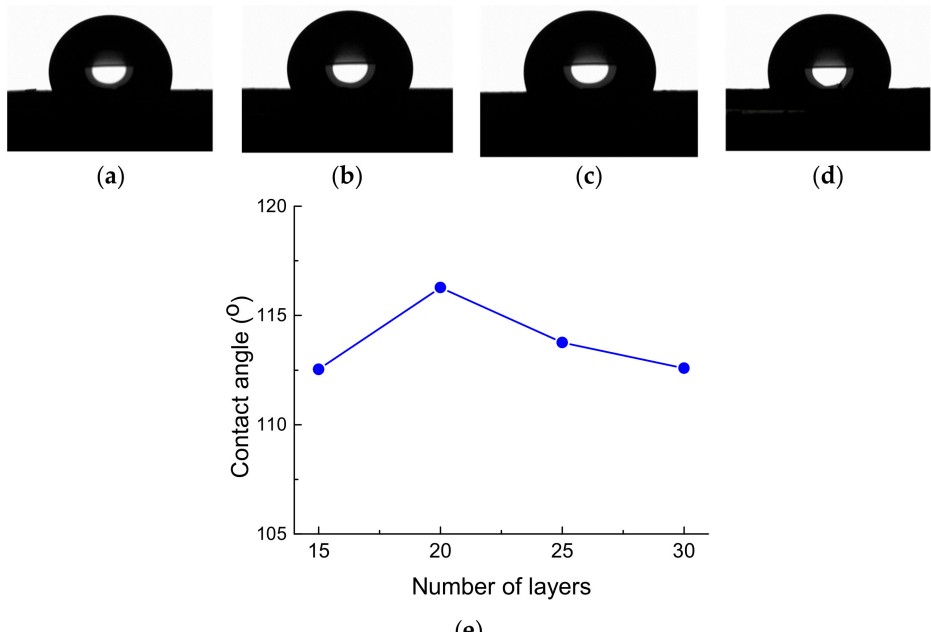

**Figure 9.** Variation of contact angle of CHA coatings with increasing number of layers (**e**) and the images of water drops on CHA surfaces coated with (**a**) 15, (**b**) 20, (**c**) 25, and (**d**) 30 sol-gel layers.

All CHA samples were immersed into simulated body fluid (SBF) for one month. The phases and morphology were examined after 2, 3, and 4 weeks. The XRD patterns of the CHA samples with 15, 20, 25 and 30 layers after immersion into SBF for 2, 3 and 4 weeks are presented in Figures 10–13. Once placed into SBF, TCP starts to dissolve and induce formation of amorphous calcium phosphate (ACP) and precipitation of CHA. However, the amount of precipitated ACP and CHA is probably too small to be detected using XRD [42]. ACP is the precursor phase of bone-like hydroxyapatite [43]. Loss of crystallinity due to lower peak intensity is visible in Figures 10 and 11 for samples with 15 and 20 layers respectively.

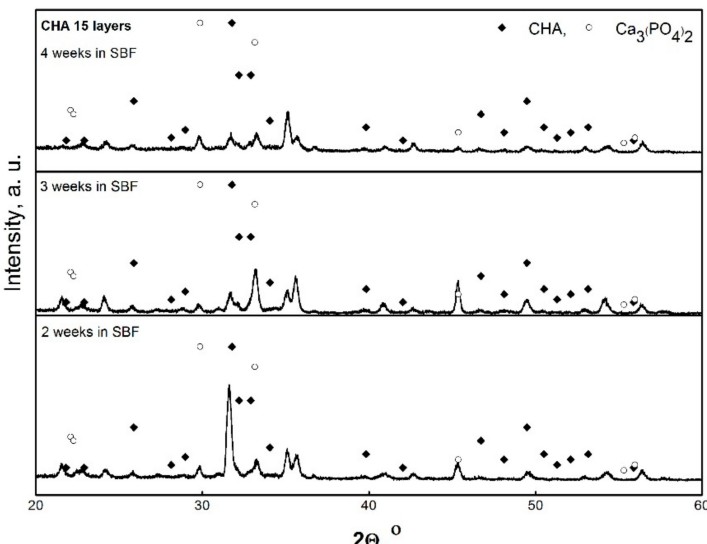

**Figure 10.** XRD patterns of CHA samples coated with 15 layers after immersion in SBF for 2, 3, and 4 weeks.

Phase changes during one month of soaking in SBF are not monotonic.

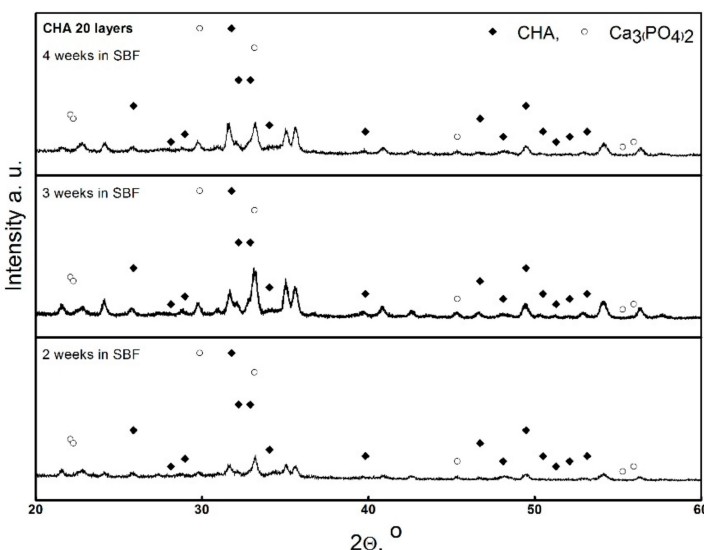

**Figure 11.** XRD patterns of CHA samples coated with 20 layers after immersion in SBF for 2, 3, and 4 weeks.

The situation changed with 25 layers (Figure 12).

After 1 month of soaking in SBF, a decrease in the intensity of the peaks attributable to both CHA and TCP phases can be seen in the XRD patterns. Amorphous precipitate of CHA and ACP dominated the surface of stainless steel. However, the presence of amorphous calcium phosphate has a very positive impact for bone repair [44]. With further increases in the number of CHA layers, the dissolution and precipitation process remains very similar (Figure 13).

Slightly different actions for similar samples in SBF solution have been observed in other studies. For example, Chen et al. determined that in the early soaking stage the dissolution action dominates [45]. Following this, new precipitates form, indicating that precipitation reaction is dominating. In addition, this process can be different when the soaking time is increased. Thus, the amorphous phosphate phase on the surface formed with increasing soaking time. The most obvious change occurs in the samples with 25 and 30 layers of CHA. Probably due to precipitation of ACP from SBF, the diffraction peaks of

CHA were no longer visible after soaking for more than 3 weeks. The XRD results clearly indicate that ACP formed, which is suitable as a bone substitute material, having sufficient cell proliferation and ALP activity [46].

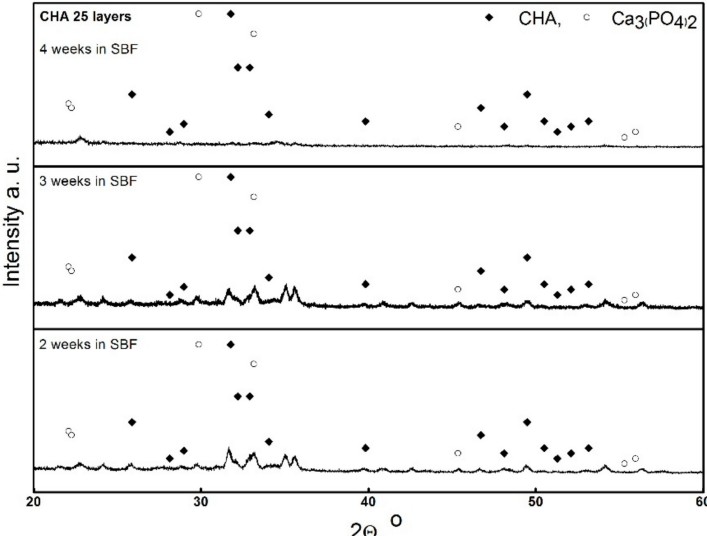

**Figure 12.** XRD patterns of CHA samples coated with 25 layers after immersion in SBF for 2, 3, and 4 weeks.

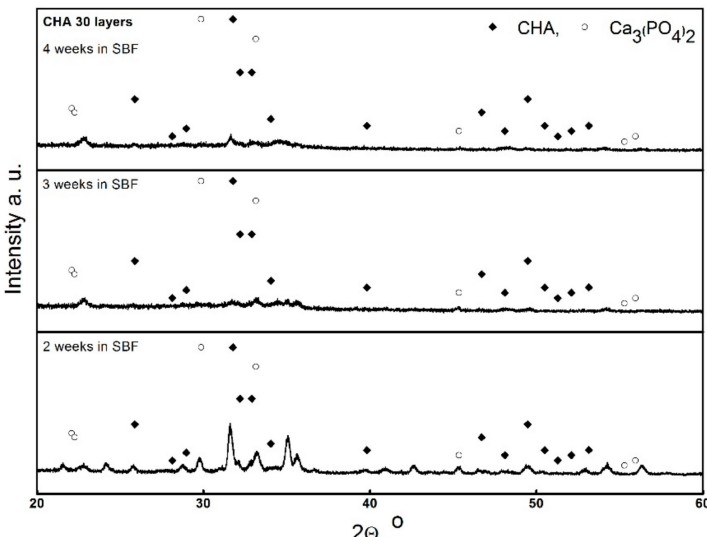

**Figure 13.** XRD patterns of CHA samples coated with 30 layers after immersion in SBF for 2, 3, and 4 weeks.

The SEM micrographs of the CHA samples with 15, 20, 25, and 30 layers obtained after immersion into SBF for 2, 3, and 4 weeks can be seen in Figure 14.

Interestingly, almost no changes in the surface morphology of CHA coatings were observed after soaking in SBF. The microstructure was not influenced by immersion time or by the number of layers on the substrate, despite the slight changes in phase composition and crystallinity during SBF immersion. Neither etching behavior, nor additional precipitates were observed. Probably, the dissolution and the precipitation processes proceeded simultaneously, without any domination [45]. However, SEM images obtained using secondary electron or backscattered electron modes give poorly distinguishable results among different calcium phosphate phases [46–49]. Loss of crystallinity and formation of various defects including twining, dislocations, stacking faults, and grain boundaries should be investigated using transmission electron microscopy [50].

The SEM results were partially confirmed by the contact angle measurements. All CHA coatings after soaking in SBF were more hydrophilic compared to the initial samples. The values of the contact angle were almost the same for all (100°–98°). Thus, wettability of the CHA samples containing 15–30 layers did not change significantly after 4 weeks of soaking in SBF.

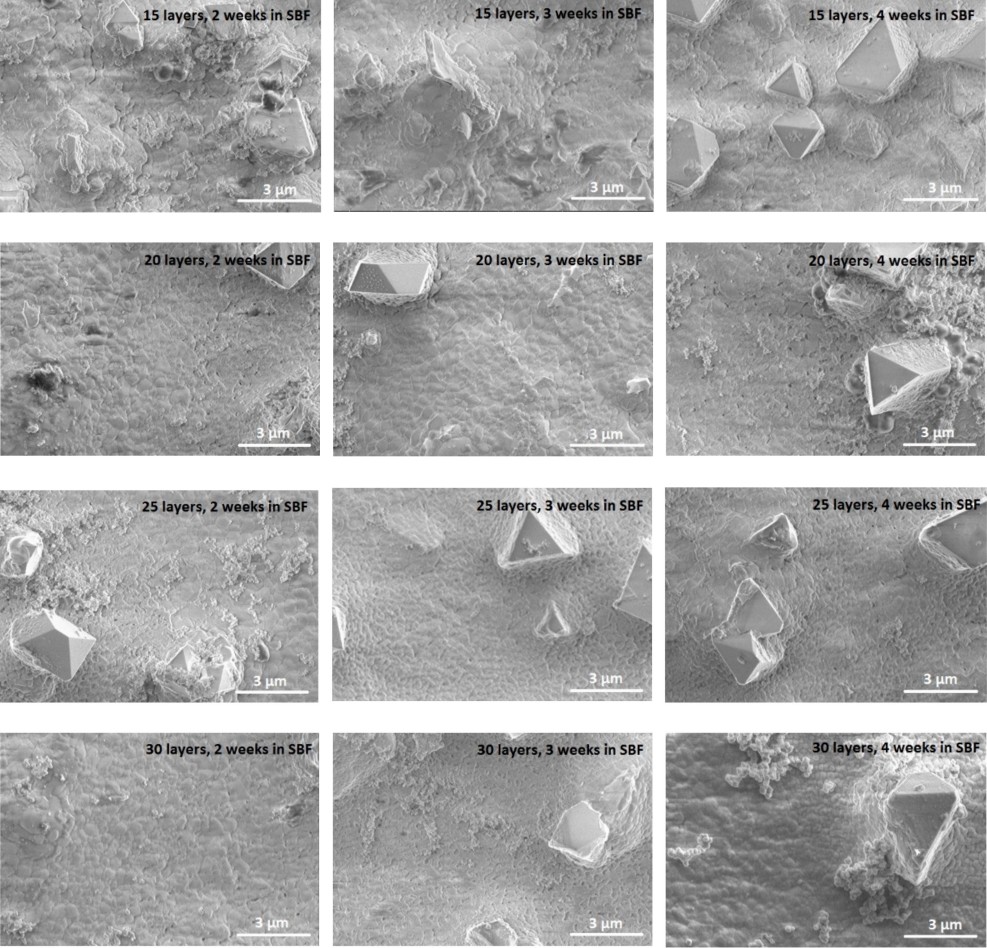

**Figure 14.** SEM micrographs of CHA samples coated with 15, 20, 25, and 30 layers after immersion in SBF for 2, 3, and 4 weeks.

## 4. Conclusions

An aqueous sol-gel method was used for the synthesis of calcium hydroxyapatite ($Ca_{10}(PO_4)_6(OH)_2$; CHA) thin films on medical grade stainless steel substrates with transverse and longitudinal patterned roughness. Each layer in the preparation of CHA multilayers (15, 20, 25, and 30) was separately annealed at 850 °C in air. According to the XRD analyses results, the most crystalline CHA sample was obtained with 25 spin-coating and annealing layers. The FTIR and Raman spectroscopy results were in good agreement with the XRD analysis data, confirming the formation of good quality CHA coatings on the rough surface of stainless steel. The formation of CHA coatings was also confirmed by XPS analysis. The chemical bonding such as P–O and Ca–O were noticed in the CHA thin films using XPS measurements. The samples with 25 layers had the lowest friction under both 1 and 5 N loads. The contact angle of coatings (113°–116°) showed the formation of hydrophobic layers of CHA. CHA coated samples were immersed in simulated body fluid (SBF) for 2, 3, and 4 weeks and formed amorphous calcium phosphate (ACP). However, the microstructure of SBF soaked CHA samples was not influenced by immersion time or by the number of layers on the substrate, despite the phase composition and crystallinity differences. All CHA coatings after soaking in SBF were more hydrophilic compared to the initial samples, which could improve ossteointegration and promote bone



cell proliferation for a better bone-implant connection. ACP is biocompatible and a precursor for CHA, indicating this coating material may be suitable to promote cell proliferation and ALP activity.

**Author Contributions:** Formal Analysis, V.J., A.Z. and A.K..; Investigation, V.J., S.S., S.W.C., R.J., A.S. and T.M.; Resources, T.C.K.Y., K.I. and R.R.; Data Curation, V.J.; Writing—Original Draft Preparation, V.J.; Writing—Review and Editing, A.K.; Visualization, V.J.; Supervision, S.W.C. and A.K.

**Funding:** This research was funded by The Japan Society for the Promotion of Science (JSPS). Fellow's ID No.: L12546.

**Conflicts of Interest:** The authors declare no conflict of interest.

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
