# Peer review of "Characterization of Sol-Gel Derived Calcium Hydroxyapatite Coatings Fabricated on Patterned Rough Stainless Steel Surface"

_coatings, doi:10.3390/coatings9050334_

Round 1
Reviewer 1 Report
1. This manuscript had interesting data for a coating technique for metal implants. The authors never related how their results would be on a metal implant?
2. The authors discussed implantation in the introduction but did not address the issues they discussed such as fibrous encapsulation.
3.The scale of the roughness was not described- microns, millimeters? Does this roughness correspond to ANY metal implant?
4. How thick was each coating and overall how thick?
5. How long was each annealing cycle?
6. List the country of manufacture for chemicals and instruments throughout.
7. Use "number of layers", not "amount of layers".
8. Use past tense for your results.
9. Particles are separate items, you created grains (which are interconnected/attached) in a thin film.
10. Are you getting dissolution or precipitation from the SBF? You didn’t show evidence of dissolution?
Be direct.
Brevity!

Author Response
1. This manuscript had interesting data for a coating technique for metal implants. The authors never related how their results would be on a metal implant?
CHAp thin films were fabricated on the stainless steel substrates. The next step of our investigations willbe preparation of prototype CHAp coatings on metal implants.
2. The authors discussed implantation in the introduction but did not address the issues they discussed such as fibrous encapsulation.
Fibrous encapsulation now is also discussed in the introduction part as reviewer suggested.
3. The scale of the roughness was not described- microns, millimeters? Does this roughness correspond to ANY metal implant?
The roughness scale was less than 0.2 micrometers. This information was added to the experimental part.
As far as we know, coated implants sometimes are rough to increase adhesion of coatings.
4. How thick was each coating and overall how thick?
The thickness of each layer was approximatelly 30 nm. Overall thickness of coatings is up to 1 micron.
5. How long was each annealing cycle?
5 hours in air. This information added to the experimental part.
6. List the country of manufacture for chemicals and instruments throughout.
Manuscript amended as requested.
7. Use "number of layers", not "amount of layers".
Manuscript amended as requested.
8. Use past tense for your results.
Manuscript amended as requested.
9. Particles are separate items, you created grains (which are interconnected/attached) in a thin film.We agree with the comment. Manuscript amended as requested.
10.Are you getting dissolution or precipitation from the SBF? You didn’t show evidence of dissolution?
In our opinion, the simultaneous dissolution-precipitation processes occurs.
11. Too indistinct to read. Do you need all 4 specta?
Font in the pictures increased. We believe that all 4 spectra are necessary in order to compare samples with different number of layers.
12. which peaks?
font increased, peaks labelled in the picture.
13. are the crystals TCP? inform the reader
in our opinion crystals are iron oxides. Explanation added to the manuscript.
14. add lavels of # of layers to each picture; make the micro markers visible!
Manuscript amended as requested.
15. put the legend in order to match the tracings to make it easier to understand.
Manuscript amended as requested.
16. what is "effectiveness"
Tribological effectiveness is the ability to reduce surface friction and maintain low COF for as many friction cycles as possible.
17: how do you define durability and get the 30 vs, 100 here- please explain
The durability shows the ability to resist friction without getting damaged of CHAp layers. CHAp layers obtained after 25 coating cycles sustain low Coefficient of Friction (COF<0.2) for 30 and 100 friction cycles under 5 N and 1 N loads respectively until it becomes to increase. The enlargement of COF value represents the gradual degradation of CHAp layers leading to higher surface friction and/or wear.
18. what kind of water? Tap? Distilled? Deionized? Why not PBS?
we used distilled water. This information was added to the experimental part.
Other changes suggested by the reviewer also applied.
Reviewer 2 Report
The manuscript looks interesting but I suggest to perform more analysis on the stability and adhesion of the coatings.
Author Response
1.The manuscript looks interesting but I suggest to perform more analysis on the stability and adhesion of the coatings.
we agree with the reviewer that adhesion is very important chacarteristic of functional coatings, however due to small sample size not all measurements are possible. At this stage we aimed to investigate chemical composition and impact of SBF. Very likely we will investigate stability and adhesion in the near future with larger samples.
Reviewer 3 Report
The authors present a study on the preparation and characterization of calcium hydroxyapatite coatings on patterned stainless steel substrates as is and after different durations in simulated body fluid. They characterize various surface, physical and chemical properties of the coatings using diffuse reflectance infrared Fourier transform spectroscopy, X-ray diffraction, Raman spectroscopy, XPS spectroscopy, scanning electron microscopy. They also perform contact angle measurements and investigate tribological properties.
1. The manuscript needs many English language improvements
2. In Line 158, do the authors mean 535.1 eV (instead of 335.1 eV as they currently have)
Author Response
1.The manuscript needs many English language improvements
English was carefully revised.
2. In Line 158, do the authors mean 535.1 eV (instead of 335.1 eV as they currently have)
thank you for noticing this typo. The value is 535.1 eV. Manuscript amended.
Round 2
Reviewer 1 Report
My comments are on the attached paper. Much improved. However, using the past tense for results is best, not present tense. See comments.
Be brief.
Use intense, not intensive.

Author Response
1.Shorter titles are better
Title amended and now is “Characterization of Sol-Gel Derived Calcium Hydroxyapatite Coatings Fabricated on Patterned Rough Stainless Steel Substrate”
2. Same or similar? (In sentence “The similar sol-gel method was used for the synthesis of CHAp coatings on stainless steel substrate“)
Same method. Sentence amended accordingly.
3.Unclear sentence “No specific features proving the formation of oxyhydroxyapatite Ca10(PO4)6(OH)2-2xOx as was detected in the 143 FTIR spectra of CHAp this films deposited on Si substrate was observed in this our case“
Sentence rewritten and now is „No specific bands attributable to oxyhydroxyapatite Ca10(PO4)6(OH)2-2xOx was detected in our spectra, however it was the case for the CHA thin films deposited on Si substrate.”
4. Technically, when your samples are analyzed, you don't have sol-gel, you have transformed the gel into an amorphous & crystalline layer at 850°C.
We agree with this comment. The incorrect term was used by mistake.
5. What are morphological features?
By writting „Morphological features“ we meant smoothness. Sentence amended for clarity and now is „The surface smoothness was better when the number of layers was 20.”
6. The test description should not be in results.
Test description transferred to experimental part.
7. What do you mean by less relevant? (“Thin layers, produced after 15 coating cycles, were tribologically effective under 5 N load while effect under 1 N load was less relevant”)
We meant that low COF was not observed under 1 N load. Sentence now is “Thin layers, produced after 15 coating cycles, were tribologically effective under 5 N load while under 1 N load this effect was not observed”.
8. Do you mean dissolution or precipitation of ACP? What is dissolving? This concept of dissolution and reprecipitation merits a brief description.
Explanation added to the text. Paragraph slightly rewritten. “Once placed into SBF TCP starts to dissolve and induce formation of amorphous calcium phosphate (ACP) and precipitation of CHA. However, the amount of precipitated ACP and CHA probably is too small to be detected by XRD. ACP is the precursor phase of bone-like hydroxyapatite.”
9. Can these techniques give distinguishable results or not? Your sentence is contradictory as written.
Sentences rewritten for clarity and now is ” However, SEM images obtained using secondary electron or backscattered electron modes give poorly distinguishable results among different calcium phosphate phases.46,48-50 Loss of crystallinity and formation of various defects including twining, dislocations, stacking faults and grain boundaries should be investigated by transmission electron microscopy.46
10.Why are you underlining some references?
This happened by mistake during editing process. Underline removed from references.